# Decontamination Strategies Used for AFB Culture Significantly Reduce the Viability of *Mycobacterium abscessus* Complex in Sputum Samples from Patients with Cystic Fibrosis

**DOI:** 10.3390/microorganisms9081597

**Published:** 2021-07-27

**Authors:** Dominic Stephenson, Audrey Perry, Andrew Nelson, Ali E. Robb, Matthew F. Thomas, Stephen J. Bourke, John D. Perry, Amanda L. Jones

**Affiliations:** 1Microbiology Department, Freeman Hospital, Newcastle upon Tyne NE7 7DN, UK; dominic.stephenson@northumbria.ac.uk (D.S.); audrey.perry@nhs.net (A.P.); ali.robb@nhs.net (A.E.R.); 2Faculty of Health and Life Sciences, Northumbria University, Newcastle upon Tyne NE1 8ST, UK; andrew3.nelson@northumbria.ac.uk (A.N.); amanda.l.jones@northumbria.ac.uk (A.L.J.); 3Paediatric Respiratory Unit, Great North Children’s Hospital, Newcastle upon Tyne NE1 4LP, UK; matthew.thomas17@nhs.net; 4Adult Cystic Fibrosis Centre, Royal Victoria Infirmary, Newcastle upon Tyne NE1 4LP, UK; stephen.bourke1@nhs.net

**Keywords:** *Mycobacterium abscessus*, *Mycobacterium avium*, cystic fibrosis, decontamination, RGM medium

## Abstract

Nontuberculous mycobacteria are important respiratory pathogens in patients with cystic fibrosis (CF). For diagnosis, international guidelines recommend culture of sputum that has been decontaminated via chemical treatment. Fifty-six sputum samples from 32 patients known to be previously colonized or infected with NTM were subdivided, and the aliquots were subjected to six different decontamination strategies, followed by quantitative culture for NTM. Thirty sputum samples contained *Mycobacterium abscessus* complex (MABSC) and 11 contained *Mycobacterium avium* complex (MAC). Decontamination strategies included treatment with N-acetyl L-cysteine with 2% sodium hydroxide (NALC-NaOH), 4% NaOH, 1% chlorhexidine, 0.5 N sulfuric acid, 5% oxalic acid, double decontamination with NALC-NaOH, followed by 5% oxalic acid, and saline (0.85%) as a control. The samples were also cultured directly with no treatment. Treatment with NALC-NaOH resulted in an average reduction in colony count of 87% for MABSC when compared with direct culture. NaOH at 4% caused a 98.3% average reduction in colony count. All treatments that included NaOH resulted in colony counts that were statistically lower than those obtained from direct culture or the saline-treated control (*p* < 0.05). Standard treatments using sulfuric or oxalic acids were less deleterious, but still resulted in an average reduction in colony count of at least 30%. The viability of MAC was much less affected by most decontamination treatments. In conclusion, the viability of MABSC was severely compromised by standard decontamination regimens. This supports recent evidence showing that optimal recovery of MABSC is achieved by culture on an appropriate selective agar without decontamination of sputum samples.

## 1. Introduction

Nontuberculous mycobacteria (NTM) are important pathogens in individuals with cystic fibrosis (CF) and other lung diseases. The dominant pathogens in this context are *Mycobacterium abscessus* complex (MABSC) and *Mycobacterium avium* complex (MAC). *Mycobacterium abscessus* has recently been reclassified as *Mycobacteroides abscessus* [1], although both names remain taxonomically valid [2]. Infection with MABSC, in particular, is associated with a severe decline in lung function and has a worse impact than infections caused by other lung pathogens, such as *Pseudomonas aeruginosa* and *Burkholderia cepacia* complex [3]. Consequently, there is a strong case for the early detection of MABSC in the respiratory tract, before infection becomes established and difficult to eradicate [4].

Diagnosis of NTM infection is achieved by culturing sputum samples using methods that were designed primarily for the isolation of *Mycobacterium tuberculosis*, which is comparatively rare in CF. Such methods are often referred to as acid-fast bacilli (AFB) culture and they involve the chemical decontamination of sputum samples to eliminate any non-mycobacteria that might otherwise overgrow *M. tuberculosis* or NTM. A wide range of decontamination strategies have been recommended in an attempt to find a balance between effective decontamination of samples and maintaining the viability of mycobacteria. For example, the UK Standards for Microbiology Investigations published by Public Health England in 2020 offer a choice of five different decontamination methods, depending on the level of contamination expected in the sample [5]. These include the use of 1 N sodium hydroxide (NaOH 4% *w*/*v*), 2% NaOH with N-acetyl-L-cysteine, sulfuric acid (0.5 N), trisodium phosphate with benzalkonium, and oxalic acid (5%) [5].

In 2016, international recommendations from The Cystic Fibrosis Foundation (CFF) and European Cystic Fibrosis Society (ECFS) advocated one method for decontamination of respiratory samples using N-acetyl-L-cysteine (NALC (0.5%)–NaOH (2%)) [6]. However, in cases where a sample remains contaminated with Gram-negative bacteria after standard NALC-NaOH decontamination, it should be further treated with either 5% oxalic acid or 1% chlorhexidine [6]. Others have routinely applied dual treatment strategies to all sputum samples from patients with CF, for example, by using NALC-NaOH followed by treatment with oxalic acid [7].

In 2014, Preece et al. described a highly selective culture medium (RGM medium) containing a novel ‘cocktail’ of antimicrobials that allows for effective culture of NTM without the requirement of chemical decontamination [8]. The medium was originally designed for the isolation of rapidly growing mycobacteria (RGM) but extended incubation allows for the isolation of slow-growing NTM. The RGM medium essentially comprises Middlebrook agar supplemented with yeast extract and oleic acid–albumin–dextrose–catalase (OADC) supplement. Its high selectivity is due to the addition of colistin (32 mg/L), amphotericin (5 mg/L), fosfomycin (400 mg/L) and C-390 (32 mg/L). It has recently been commercialized as NTM Elite agar.

In this study, we aimed to utilize the RGM medium as a novel tool to allow us to quantify the impact of various methods of chemical decontamination on the viability of NTM in sputum samples from patients with CF. 

## 2. Materials and Methods

### 2.1. Patient Samples and Reagents

Unless otherwise stated, all chemicals and reagents were obtained from Sigma-Aldrich, Poole, UK. RGM medium was prepared as previously described [9]. Over an eight-month period, 56 sputum samples were referred to the Freeman Hospital Microbiology Department from 32 patients who were known to be infected or colonized with NTM. The specimens were collected as part of routine monitoring of the patients and no specimens were collected for the purposes of this study. As part of the routine processing, samples were digested with an equal volume of sputasol (Product SR0233A; Oxoid, Basingstoke, UK) and mixed thoroughly using a vortex mixer until homogeneous. They were then cultured for a range of pathogens, in line with routine laboratory procedures. Leftover aliquots of these samples were then anonymized by laboratory staff and processed, as described in Section 2.2.

### 2.2. Decontamination and Culture on RGM Medium

From each sample, 7 × 300 µL aliquots of digested sputum were transferred into sterile 2 mL screw top micro-centrifuge tubes, and each was subjected to a separate decontamination treatment, as detailed in Table 1. A further 300 µL of untreated digested sputum was inoculated onto three RGM plates (100 µL per plate). Decontaminated samples were neutralized with an excess of 0.067 M phosphate buffer (containing 0.2% phenol red; pH 6.8), and the pH was adjusted to 6.8 ± 0.2 if necessary, using either 4% NaOH or 1 M hydrochloric acid. Neutralized samples were centrifuged (3000× *g*, 15 min) and their supernatant was carefully discarded. Deposits were re-suspended in 300 µL PBS (pH 6.8) and then inoculated onto three plates of RGM medium (100 µL per plate). All plates were incubated at 30 °C and colony counts taken after 4, 7 and 14 days for MABSC, or 7, 14 and 21 days for samples containing MAC.

### 2.3. Identification of Mycobacteria

The identification of all colony-types was confirmed by matrix-assisted laser desorption ionization-time of flight mass spectrometry (MALDI-TOF MS) using a Bruker Biotyper with the in-house extraction method. Each colony type was suspended in 0.5 mL of sterile deionized water in a 2 mL centrifuge tube to make a turbid suspension, and 1 mL of pure ethanol was added to the tube. The suspension was vortexed at high speed for 2 min and then centrifuged for 2 min at 13,000 rpm. The supernatant was then carefully removed with a pipette. The tube was then re-centrifuged for 30 s and any remaining ethanol removed. Ten sterile glass beads were added to the tube containing the pellet and 50 µL of acetonitrile was added. The tube was then vortexed at high speed for 1 min. A 50 µL aliquot of 70% formic acid was then added and the tube was vortexed for a further 5 s. The tube was then centrifuged for 2 min at 13,000 rpm. Duplicate aliquots of 1.5 µL of supernatent were spotted onto a MALDI stainless steel target and both spots were covered with 1.5 µL of HCCA matrix (Bruker, Coventry, UK) and allowed to dry. Species identification was achieved by matching spectra to those available in the general database and mycobacteria were also matched to the Bruker Mycobacteria Library version 4.0.

### 2.4. Data Analysis

A one-way analysis of variance (ANOVA) was used to determine whether there were any statistically significant differences between the mean counts derived from the different treatments. We also performed post hoc analysis using Tukey HSD.

## 3. Results

Forty-one samples yielded either MABSC (*n* = 30) or MAC (*n* = 11). The other 15 samples did not yield NTM and were excluded from further analysis. The 41 samples were derived from 18 patients with CF and three patients with other lung diseases. Their age range was 10–69 years. All other demographic information was excluded by the anonymization process. Recovery of NTM following direct inoculation onto RGM medium produced the highest mean counts for both MABSC and MAC. Treatment with saline generated mean counts that were only slightly lower: 1.8% lower for MABSC and 3.3% lower for MAC (*p* = 0.95). This demonstrated that any losses of NTM during centrifugation and removal of supernatant were negligible. As direct inoculation enabled the greatest overall recovery, the impact of decontamination was expressed as a percentage of this count. For the purposes of this calculation, it was assumed that the count on RGM represented a 100% recovery, although this cannot be proven.

### 3.1. Impact of Decontamination on the Recovery of MABSC

Figure 1 shows the impact of the various decontamination treatments on the recovery of MABSC from sputum samples. The recovery of MABSC was affected most severely by the treatments that incorporated sodium hydroxide with an 87% reduction in colony-forming units (CFU) caused by NALC with 2% sodium hydroxide, increasing to a 98.3% reduction when using 4% sodium hydroxide. There was a statistically significant difference between group means at the *p* < 0.05 level by one-way ANOVA (F(7, 232) = 4.020, *p* = 0.0004). Post hoc analysis using Tukey HSD showed that average counts following any of the three treatments incorporating sodium hydroxide were significantly lower than counts resulting from direct inoculation or from control samples processed with saline. Mean counts were also substantially reduced when using sulfuric acid (37.6%), oxalic acid (32.2%) and chlorhexidine (56.1%), but these reductions were not statistically significant. Figure 2 shows an example of the impact of NALC-NaOH on the recovery of MABSC.

### 3.2. Impact of Decontamination on the Recovery of MAC

Figure 1 also shows the impact of the various decontamination treatments on the recovery of MAC from sputum samples. For MAC, there was no statistically significant difference between group means at the *p* < 0.05 level by one-way ANOVA (F(7, 80) = 0.326, *p* = 0.940). Although not significant, all decontamination treatments negatively impacted MAC recovery compared to either saline treatment or direct inoculation methods. Oxalic acid treatment was least detrimental, causing only a 15.6% reduction in count, whereas chlorhexidine was most detrimental, causing, on average, a 66.7% reduction in count.

## 4. Discussion

The deleterious impact of decontamination on the viability of mycobacteria has been recognized for many years, and several authors have recommended the use of selective agars and the avoidance of chemical decontamination of samples. For example, 40 years ago, Rothlauf et al. compared a selective Middlebrook 7H10 medium with a traditional decontamination method utilizing NALC-2% NaOH and non-selective 7H11 medium [13]. In a large study with 10,782 samples, they reported that contamination was more than threefold lower using the selective medium, and only six mycobacteria were not recovered on the selective medium compared with 61 that were not recovered following decontamination. Despite this, others did not report the same success with selective media [14], and the chemical decontamination of samples has remained a standard method for the isolation of mycobacteria. This had led to the examination of many different approaches to decontamination to try to reduce contamination to an acceptable level whilst not killing off the mycobacteria in the samples [10,11,12,13,14,15,16,17,18].

The effective recovery of NTM from sputum is particularly challenging in the context of CF. This is due to the frequent presence of high levels of *Pseudomonas aeruginosa*, other Gram-negative bacteria (e.g., *Achromobacter*, *Burkholderia*) and fungi that can frequently contaminate and overgrow mycobacterial cultures. Whittier et al. proposed a dual decontamination strategy using NALC-NaOH followed by 5% oxalic acid to reduce the contamination rate to an acceptable level [15]. This approach was endorsed by Bange and Böttger who reported that a high contamination rate of 45.1% could be reduced to an acceptable rate of 7.3% with subsequent use of oxalic acid [11]. Later, Ferroni et al. compared the use of 1% chlorhexidine with the use of NALC-NaOH-Oxalic acid with 827 sputum samples from patients with CF [12]. They reported that the use of 1% chlorhexidine allowed for the recovery of twice as many NTM (*p* < 0.0001), despite a higher contamination rate. Notably, of the 43 specimens containing MABSC, 42 (98%) were detected after decontamination with 1% chlorhexidine, compared with only 21 (49%) using NALC-NaOH-Oxalic acid [12]. These findings were endorsed by De Bel et al., who detected NTM in 15 of 18 positive sputum samples after chlorhexidine treatment compared with only 5 out of 18 when using NALC-NaOH-Oxalic acid, when solid media were used for culture [18]. Finally, Jordan et al. seeded sterilized sputum with known quantities of MABSC and then compared three different decontamination treatments to examine their impact on viability [10]. They reported that by using either 3% oxalic acid or NALC-2% NaOH- 5% Oxalic acid, they could detect 760 CFU/mL, whereas the use of 4% NaOH required an inoculum of 11,000 CFU/mL, suggesting that 4% NaOH severely reduced the viability of MABSC.

RGM medium is a highly selective medium designed for the isolation of NTM [9], and we have utilized this medium to quantify the impact of various decontamination strategies. The most notable finding is the substantial negative impact of NaOH on the viability of MABSC. According to guidelines published by Public Health England, “decontamination of specimens using NALC-NaOH is a preferred method for the digestion step because it is the least toxic to mycobacteria and, therefore, provides the highest yield of positives. It is the most commonly used method in clinical laboratories” [3]. Furthermore, decontamination using NALC-NaOH is recommended in international guidelines for processing sputum from patients with CF [6]. Despite these recommendations, there are important limitations to the choice of NALC-NaOH. Firstly, if this is used as the sole decontamination agent, it is often unsuccessful for CF sputum samples; as previously mentioned, Bange and Böttger reported a high contamination rate of 45.1% [11]. Secondly, as we report here, there is an 87% mean reduction in the count of MABSC caused by NALC-NaOH.

One important consideration is whether the impact of decontamination on the viability reported here and elsewhere [10] is reflected by the failure to detect MABSC in clinical practice. Four studies have compared the isolation of NTM following a decontamination strategy that included use of NaOH with direct culture on RGM for 28 days (without decontamination). Table 2 summarizes the isolation rates for MABSC in these four studies. In all of these studies, the isolation rate of MABSC was highest when decontamination was avoided, and in three of these studies, this difference was significant (*p* < 0.05). In the largest of these studies [19], it was shown that for 16/16 samples that contained low levels of MABSC (≤200 CFU/mL of sputum, as determined by culture on RGM medium), none could be detected using AFB culture.

With respect to MAC, there was a lower impact of the various decontamination treatments examined here, with NALC-NaOH reducing the viable count by 28.9%. This is consistent with the fact that there is little, if any, advantage of using RGM medium for isolation of MAC when compared to AFB culture, as documented in previous studies [7,19,20,21]. A limitation of this paper is that we are not able to offer conclusions on NTMs other than MABSC and MAC.

## 5. Conclusions

We have shown that all decontamination treatments reduce the viability of both MABSC and MAC in sputum samples. This effect is particularly acute for treatments containing NaOH against MABSC. This supports the abundance of recently acquired evidence showing that the optimal recovery of MABSC is achieved by culture on an appropriate selective agar, without decontamination of sputum samples [7,19,21].

## Figures and Tables

**Figure 1 microorganisms-09-01597-f001:**
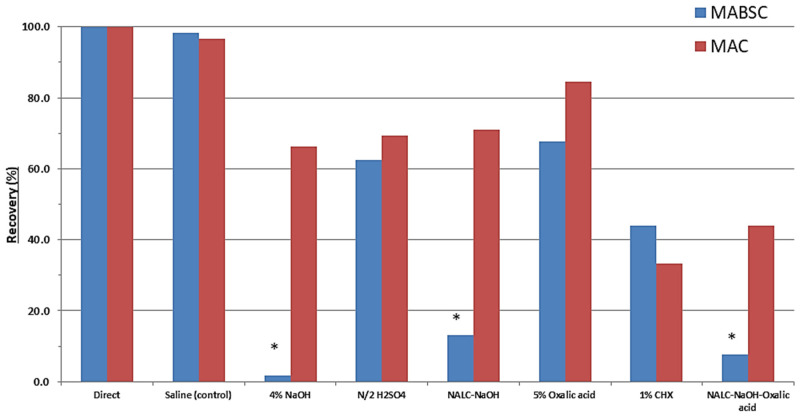
Mean % recovery of MABSC (*n* = 30; blue bars) and MAC (*n* = 11; red bars) in sputum samples caused by various decontamination treatments, compared to direct inoculation. (* *p* < 0.05).

**Figure 2 microorganisms-09-01597-f002:**
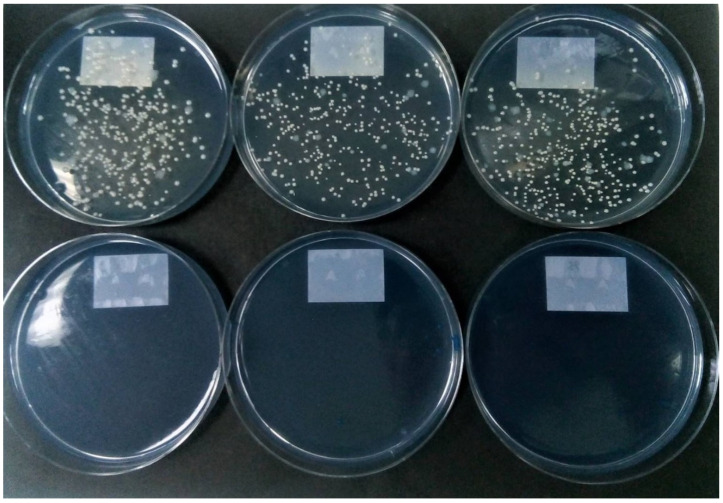
Top row: Replicate cultures of MABSC from sputum showing rough and smooth colony variants recovered by direct inoculation of RGM medium (100 µL of untreated digested sputum per plate). Bottom row: replicate cultures of the same sputum sample on RGM medium showing no colonies following treatment with NALC-NaOH.

**Table 1 microorganisms-09-01597-t001:** Reagents and conditions used for sample decontamination.

Treatment	Sample: Decontaminant Ratio	Treatment Duration	Reference for Method
0.85% Saline (control)	1:1	15 min	-
4% NaOH (modified Petroff method)	1:1	25 min	[10]
0.5 N H_2_SO_4_	1:1	60 min	[5] *
NALC + NaOH (2%)	1:1	15 min	[5]
5% Oxalic acid	1:1	15 min	[11]
1% Chlorhexidine (CHX)	1:3	15 min	[12]
Dual treatment with NALC-NaOH (2%) + 5% Oxalic acid	1:1 & 1:1	15 min & 15 min	[11]

* For treatment with 0.5 N sulfuric acid, the guideline states that the duration of decontamination should be reviewed in light of individual laboratory contamination rates for different specimen types. Internal validation studies led us to select a duration of 1 h.

**Table 2 microorganisms-09-01597-t002:** A comparison of direct culture on RGM medium with AFB culture of decontaminated samples for isolation of MABSC from respiratory samples.

Study [Reference]	Sensitivity (%)			Details of Comparator Method	
(no. of Samples)	RGM	Comparator	*p*	Decontamination	Culture Media
Plongla et al. [7]	92.2	47.1	<0.0001	NALC-2% NaOH	MGIT + LJ medium
(*n* = 212)				+5% oxalic acid	
Stephenson et al. [19]	96.1	58.8	<0.0001	4% NaOH	MGIT + LJ medium
(*n* = 1002)				0.5 N H_2_SO_4_	MGIT + LJ medium
Rotcheewaphan et al. [20]	96.4	89.3	0.6	NALC-2% NaOH	MGIT+
(*n* = 203)					Middlebrook 7H11
Brown-Elliott et al. [21]	94	66	<0.05	NALC-2% NaOH	Middlebrook 7H11
(*n* = 297)				+/−5% oxalic acid	Mitchison Selective agar
“ ” “ ” “ ” “	94	61	<0.05	“ ” “ “ ”	Versatrek broth

Abbreviations: MGIT, mycobacterial growth indicator tube; LJ medium: Löwenstein–Jensen medium.

## Data Availability

All relevant data is included in the manuscript.

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
