# Peer review of "Decontamination Strategies Used for AFB Culture Significantly Reduce the Viability of Mycobacterium abscessus Complex in Sputum Samples from Patients with Cystic Fibrosis"

_microorganisms, 2021, doi:10.3390/microorganisms9081597_

Round 1

Reviewer 1 Report

This is a highly significant and well designed contribution to the operation of clinical microbiology laboratories.

The following points should be addressed.

The reductions (line 131 etc) being not statistically significant is puzzling. Please provide an explanation.

Please provide a reference for the MALDI-TOF method for identifying the colony types.

Please separate 0.5 or 1 from "N". Also, 0.5 N is preferable to N/2

RGM medium does not appear to be available commercially. Please provide its composition.

In Table 1, please reduce the space between "decontaminant" and "ratio", change the font of the title of the last column, and use 1:1 instead of 1 : 1.

Author Response

Please see the attached file entitled "Response to reviewers"

Reviewer 2 Report

I found the article an interesting reading. I just have minor comments:

  • RGM medium: I think the readers would benefit from having more information about this medium. For instance, the abbreviation stands for, it is based on what, what makes it selective, etc.
  • Figure 1: Error bars should be added to the graph.

Author Response

Please see the attached file entitled "Response to reviewers".
